# Mediation and moderation effects of health system structure and process on the quality of mental health services in Ghana – structural equation modelling.

Eric Badu[1]*, Anthony Paul O'Brien[2], Rebecca Mitchell[3], Akwasi Osei[4]

**1** School of Nursing and Midwifery, Faculty Health and Medicine, The University of Newcastle, Newcastle, Australia, **2** School Nursing and Midwifery, Faculty Health and Medicine, University of Newcastle, Newcastle, Australia, **3** Macquarie Business School, Macquarie University, Sydney, Australia, **4** Ghana Health Services, Accra, Ghana

* Eric.badu@uon.edu.au, badu3eric@gmail.com

## Abstract

### Introduction

Incorporating consumers' perspectives into the quality of mental health service measurement is a growing priority among mental health professionals' and policymakers. Despite this, there is limited empirical evidence related to consumer perspectives of quality of mental health services. This study, therefore, aims to measure the mediation and moderation effects of health system structure and process on mental health quality in Ghana.

### Methods

A random sample of 510 consumers were recruited to complete the Verona Satisfaction Scale (54-items), together with the WHO Disability Assessment Instrument (36 items) using the Redcap application. Confirmatory factor analysis (CFA) and Structural Equation Modelling were used to test the hypothesised theory using STATA 15.

### Results

The CFA showed that the hypothesised model had a good fit to the data. The findings confirmed the hypothesis that the *process* constructs mediate the relationship between the health system *structure* and the *outcome* of mental health services. Specifically, the health system *structure* had a positive and significant causal relationship with the mediator-*process* (β = 0.60; p<0.01) and *outcome* (β = 0.47; p<0.01). Additionally, the mediator-*process* had a positive causal relationship with the *outcome* of the mental health services (β = 0.32; p<0.01). Insurance status (β = 0.07; p>0.05) and type of services (β = 0.025; p>0.05) had a positive moderating effect on the relationship between health system *structure* and *outcome* but were not significant.

**Data Availability Statement:** All relevant data are within the paper and its Supporting Information files

**Funding:** This work was supported by the University of Newcastle Graduate Research Australia Doctoral Scholarship to EB. The funder had no role in study design, data collection and analysis, decision to publish, or preparation of the manuscript.

**Competing interests:** The authors have declared that no competing interests exist.

## Conclusion

Improvements to mental health system structure and the process could promote the quality of services as experienced by consumers. Government stakeholders are encouraged to accordingly strengthen health systems with the aim of improving the mental health outcomes for consumers.

## Introduction

The quality of mental health services generally describes the degree to which services for consumers and populations increase the likelihood of desired outcomes and are consistent with current professional knowledge and practice [1, 2]. Several strategies have been proposed to improve the quality of mental health services as experienced by consumers. For example, several high-income countries have developed national strategic approaches to mental health reform with quality initiatives to achieve clinical improvements [2–5]. Specifically, quality checklists, glossaries, and documentation of health policy, programs, and institutional quality assurance reviews have been implemented by professional and government agencies to determine, monitor and improve the level of quality of services [2, 3]. The Organisation for Economic Co-operation and Development, as well as International Organisation for Standardisation, have recently promulgated quality assurance indicators to assist countries in assessing their health service outcomes, including those of mental health services [3, 6, 7]. These quality indicators have provided a firm foundation to engage governments, healthcare organisations and clinicians in the ongoing quest for quality measurement and improvement [3, 4].

The efforts towards understanding and improving the quality of mental health services delivery are often focused on issues of data source, segmentation, and thresholds [5, 8]. These thresholds generally focus on stakeholders, like the public or society, mental health professionals, as well as consumers and family caregivers [8]. The quality of mental health services from the public perspective measures the investment of the public in mental health services (eg. taxation and financing) against long-run improvement in antisocial behaviour related to aggression, begging and shouting at voices among consumers [2, 8]. Mental health professional perspectives of quality tend to focus on perception regarding the strengths and weaknesses of the health system as well as the recovery of consumers [8, 9].

Conversely, consumers and family caregivers' perspectives of mental health service quality are usually centred on their subjective experiences like the quality of life and satisfaction [2, 10]. This method of measuring the quality of mental health services focuses on holistic recovery, the facilitation of consumer social inclusion, as well as reduction of stigmatization [2, 10].

Although some studies have suggested the need to incorporate consumers' perspectives into mental health quality assessment [10], there is little evidence to suggest that this is occurring successful, particularly into clinical practice. Research evidence on mental health quality assessment has been limited to service providers, particularly in developed countries (eg, Sweden [11], USA [12].

To date, mental health research in Ghana has focused on issues such as health system weakness, enablers and barriers to accessing services [13, 14], the burden of family caregiving [15–17] and treatment pathways [18, 19]. Despite this increasing evidence, none of these studies have attempted to measure the quality of mental health services from a consumer perspective. There is a gap in the literature regarding how health system *structure* and *process* impact the perceived *outcome* of mental health services. To address this gap, a quantitative study using

structural equation modelling was undertaken to assess the mediation and moderation effects of health system *structure* and *process* on the *outcome* of mental health services in Ghana.

## Mental health quality theory underpinning the structural equation model

Both the modified Donabedian theory for measuring the quality of mental health services and the Pechansky and Thomas theory on access were used to hypothesise a model which was investigated using structural equation modelling. These theories have identified three constructs relating to service quality including health systems *structure*, *process* (technical competency and therapeutic relationship) and the *outcome* of the services [2, 10]. The health systems *structure* aligns with the Thomas and Pechansky theory on access [20]. This construct is categorized into the availability, accessibility, acceptability, affordability, adequacy (accommodation) and awareness about mental health services [2, 10, 20]. The health systems structure also considers individual factors of consumers, including demographic, socio-economics, medical history as well as belief and attitudes towards health services [10]. The process construct describes the technical competency and the therapeutic relationship between consumers and service providers [10]. The therapeutic relationship describes the communication and dissemination of information, respect and dignity of consumers, as well as an adequate sense of security and the provision of relevant information for consumers [10]. The outcome construct also describes clinical and consumer perspectives of quality mental health services. The clinical outcome reflects an objective measure, which includes the symptomatology, recovery (mental health system reports), admission and readmission rates [10]. Assessment of consumer perspective includes satisfaction, quality of life, symptoms, functioning, personal recovery, physical health-related knowledge, coping abilities, family caregiving burden, participation in economic, social, education and political activities [10, 21].

The interconnection of health systems structure, process and outcome constructs have been theorized to explain the quality of mental health services [22]. In this model, the health systems structure and process serve as proxies for measuring the outcome of mental health services, directly or indirectly [2, 11, 23]. For instance, Kilbourne, Beck [22] argued that an adequate structure (stronger health system) could provide the necessary infrastructure and equipment for reporting on key processes to monitor and improve the quality of mental health services. However, one outcome of interest (e.g. satisfaction) may not be sufficient to assess the quality of mental health services. This requires measuring the effects of the health system structure and process on mental health quality from a consumer perspective [10].

## Materials and methods

### Study setting

This cross-sectional design applied a multi-stage sampling (two-stage sampling) approach to select two psychiatric hospitals and one psychiatric unit in Ghana from July 2019 to November 2019. The study was approved by the Human Research Ethics Committee, University of Newcastle (Approval No.: H-2019-0082) as well as the Ethical Review Committee of the Ghana Health Services (Approval No: GHS-ERC 003/07/19). The primary sampling purposively selected two psychiatric hospitals, as well as one psychiatric unit in the general hospital. The two psychiatric hospitals were located in the Greater Accra region and the psychiatric unit was located in the Ashanti Region. The two psychiatric hospitals were purposively selected because they were among the only three public facilities specifically designated for psychiatric services. The psychiatric unit was purposively selected to understand the variation in mental health services at different levels and to provide information on how mental health services function at the secondary level. The location of this tertiary level hospital captures a range of consumers

from Southern and Northern Ghana in terms of, for example, geo-political demography and ethnicity. The psychiatric hospital provided outpatient and in-patient services such as medication management, case management services, intensive community treatment, forensic, rehabilitation and psychological services [14]. The facilities operated on the basis of shared goal of treatment, philosophy of care and expectation for consumers. Within secondary psychiatric unit, systematic sampling was applied to identify and recruit consumers across the selected facilities [24, 25].

## Study participants and recruitment

The participants recruited were consumers who were attending a clinical review at the outpatient or inpatient department (e.g. on admission or discharged in the last six months preceding the study). The consumers recruited were18 years and above, who had been previously assessed by a Mental Health Professional (e.g. clinical psychologist, or psychiatrist) and had self-reported, or been reported by proxy, as having a mental illness. All participants were cognitively able to answer the questionnaire. The exclusion criteria for the participants included being below 18 years, having a severe condition which could cause danger to self or others, and being cognitively unable to answer the questionnaire.

The recruitment of participants followed several steps [26]. Firstly, average daily outpatient, as well as the total number of inpatient were obtained from the records department to calculate the proportion of consumers in each of the selected facilities. The total average of daily outpatient attendance for each of the facilities was divided by the target number of participants to establish a sample interval. Secondly, background information was extracted for all the consumers (e.g. age, mental condition and severity) who had been receiving outpatient, or inpatient mental health services, for the year preceding the study. This information helped to identify the recent Mental Status Examination (MSE) results for each consumer, and further determined those who were cognitively well enough to participate in the study [26]. Within this sampling frame, every fourth consumer who had received mental health services from the respective psychiatric hospital was selected. This exercise was repeated consecutively in all the selected facilities. All eligible prospective consumer participants were then invited to voluntarily participate in the survey. The invitation package contained a letter, consent form and participant information sheet which described the purpose of the research, selection criteria, benefits of participation, confidentiality, risk, and the voluntary nature of participation [27]. Telephone numbers of the researcher, academic supervisors, as well as the Human Research Ethics Committee (HREC) of the University of Newcastle and Ghana Health Services were listed in the letter for further inquiries. A total of 582 consumers of mental health services were invited or approached across all the psychiatric facilities. Forty-eight invitees did not accept the invitation. A total of 510 consumers were recruited in the survey.

The head of each of the psychiatric facilities sent out a memo describing the study to all departments (outpatient and inpatient wards). The memo explained the research objectives, participant inclusion, and the recruitment process which was advertised in each ward and department. Additionally, within each of the psychiatric facilities, the nursing manager or supervisor introduced the researcher to consumers each morning at the outpatient and inpatient ward. This helped to gain the confidence of mental health nurses and psychiatrist who work directly with consumers and also ensured that the presence of the researcher was not unexpected or intrusive [26, 28, 29]. The participants provided a suitable date and time for the administration of the questionnaire. Before their participation, the psychologist or mental health nurse re-assessed their Mental Status and affirmed that the consumer was not a danger to self, or others, and was able to voluntarily provide their consent to participate.

## Sample size

The sample size for the study was estimated using the Cochran's [30, 31] sample size formula ($n_0 = \frac{Z^2 * (p)(q)}{d^2}$). The sample size was estimated on the assumption that 50% of consumers would seek mental health services. Using a significance level of 0.05 and degrees of freedom of 0.5 resulted in a sample size of 384.16. Also, allowing a 10% non-response rate, and a design effect of 1.5, resulted in a maximum sample size of 645 participants. The study finally recruited 510 consumers.

## Measures

**Intermediate Verona Service Satisfaction Scale (VSSS-54).** The VSSS-54 instrument is a validated, multidimensional scale used to measure the quality of mental health services from consumer perspectives [32–34]. The instrument is designed to assess the quality of mental health services delivered by a multidisciplinary team of psychiatrists, psychologists, social workers and nurses. The instrument is relevant across a broad array of both medical and psychiatric settings. The contents of the instrument covers variables that have been identified from theories, concepts and empirical evidence from mental health quality assessment [32, 33, 35–39]. The VSSS-54 covers seven domains with different items (overall satisfaction = 3 items, professionals' skills and behaviour = 24 items; information = 3 items; access = 2 items; efficacy = 8 items; types of intervention = 17 items; and, relative's involvement = 6 items [32, 33, 36]. Previous studies found that the VSSS-54 had good internal consistency (α = 0.96) [34].

**Disability Assessment Schedule (WHO/DAS).** The World Health Organization Disability Assessment Schedule was used to assess the functionality of consumers. The instrument asks about difficulties that consumers have experienced due to any health conditions which may include diseases or illnesses, or other health problems that may be short or long-lasting, injuries, mental, or emotional problems; and problems with alcohol or drugs. Specifically, the WHODAS instrument asks about how much difficulty a consumer experienced in doing certain activities. The instrument has 36 items and covers six domains, which include cognition (understanding & communicating), mobility (moving & getting around), self-care (hygiene, dressing, eating & staying alone), getting along (interacting with other people), life activities–domestic responsibilities, leisure, work & school) and participation (joining in community activities [39]. Previous research has tested that the instrument had very good to excellent internal consistency (ranged from α = 0.82 to 0.98) [40]. The instrument consists of a five-point Likert scale including none, mild, moderate, severe and extreme, or cannot do at all.

**Demographic and treatment pathways.** Demographic information and issues related to the treatment pathways were collected using a self-developed questionnaire. The demographic information included variables such as age, gender, education, marital status, primary occupation, religion, geographic location, insurance status. Also, the variables measuring treatment pathways were the mental health services consumers received, type of health facility providing the services as well as the first point of seeking mental health services.

## Data collection

The items were loaded into a smartphone using the REDCap (Research Electronic Data Capture) application [41], which appears to be the most rapid and effective way of collecting and reporting primary quantitative data. The REDCap electronic data capture tool was hosted at Hunter Medical Research Institute (HMRI), University of Newcastle. REDCap is a secure, web-based software platform designed to support data collection in empirical research studies, which helps to provide an intuitive interface for validated data collection and enables

automated data export to common statistical packages [42, 43]. Consumers who agreed to participate in the survey were invited to sign a written informed consent form before their participation [44]. In an attempt to respect the rights and integrity of participants [45], each potential participant was informed of the aims, methods, anticipated benefits and potential hazards of the study and the discomforts before data collection. On each day, the researcher handed the smartphone to each consumer, in the presence of their family caregivers when possible. Consumers read the survey items on the smartphone and ticked their response (Additional information 1). The family caregivers assisted the participants when they requested assistance. The completion of the instruments took approximately 30 to 40 minutes.

## Data analysis

The study used descriptive and inferential statistics to analyse the data. First, the mean response for each item and hypothesized dimension was computed. Confirmatory Factor Analysis (CFA) as a component of Structural Equation Modelling (SEM) was used to inferentially analyse the data. SEM describes statistical modelling that used to test the validity and reliability of substantive measurement theories. [46–48]. In doing this, our analysis followed several steps. Firstly, we identified the missing values in the dataset (standard threshold <10%). The normality of the dataset was assessed through shape, skewness, and kurtosis. The data were normally distributed. Homoscedasticity was assessed to determine whether there was consistent variance across different levels of the variable. The Variable Inflation Factor for each independent variable was tested to determine the multicollinearity. We found VIF value of less than 3 for all variables indicating that multicollinearity was unlikely to present major validity issues [46–48].

The next stage of the analysis involved factor analysis to determine factor scale structure, as well as the underlying set of variables [46, 47]. Convergent validity was measured to determine how the domains of the VSSS-54 and WHODAS variables inter-correlated [46, 48]. Reliability testing was computed using Cronbach's alpha for each domain (Cronbach's alpha, α > 0.7). The next step involved testing the discriminant validity to understand the extent to which the domains of VSSS-54 were distinct or uncorrelated with the WHODAS [46, 48]. Then Confirmatory Factor Analysis (CFA) of the hypothesised measurement theory was conducted (Fig 1). The metrics used to determine goodness of fit for the CFA were Chi-Square/df (<3 good; <5 permissible), the p-value for the model α >0.05, CFI > = 0.95, GFI >0.95, AGFI >0.80, RMSEA < 0.05, PCLOSE > 0.05 and Upper <0.10 [46–48]. A path analysis diagram was applied to test the mediation effects of *process* on the relationship between the health systems *structure* and *outcome* of mental health services [46, 48]. Finally, a path analysis diagram was generated in order to test the moderating effects of individual factors on the relationship between the health systems *structure* and *outcome* of mental health services. All statistical analyses were estimated using STATA 15.1

## Results

### Demographic characteristics

The study recruited 510 consumers of mental health services. The average age of participants was 34 years. About 51.6% were males. More than a third (40% and 35%) were educated up to senior high school and tertiary level respectively. The majority of consumers (70.9%) were single. The majority (91.9%) of consumers were Christians, 68.4% enrolled into national health insurance and 64.5% were urban residents. More than a third (46.6%) of consumers had a DSMV diagnosis of Schizophrenia, 16.4% Bipolar Affective Disorder, 16.7%, Depression and 4.9% had Co-morbid conditions (Table 1).

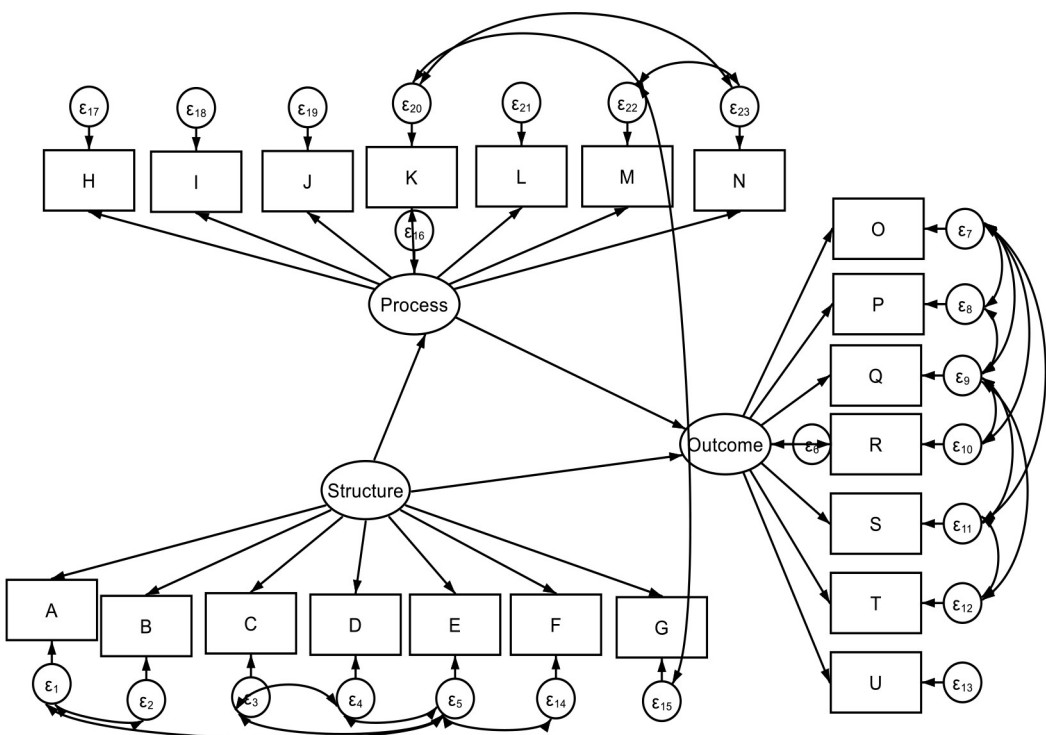

**Fig 1. Hypothesised structural equation model.**

## Descriptive statistics of items

Table 2 illustrates the mean score, standard deviation, internal consistency and factor scale for the health systems *structure*, *process* and *outcome* construct. The total reliability coefficient for the VSSS-54 items was α = 0.94 indicating excellent internal consistency. Specifically, the internal consistency of the items used for the SEM was 0.91 (health systems *structure* α = 0.75, *process* α = 0.80 and *outcome* α = 0.88). The absolute value of skewness ranged between 0.00 and 0.32 whilst kurtosis was between 0.00 and 0.38 with an acceptable threshold of +2 and—2 [46, 48]. This result indicated that the normality of the items was acceptable. The items for the SEM were identified through factor analysis. The factor analysis retained three factors with Eigenvalues of greater than two (factor one = 13.7; factor two = 4.5 and factor three = 2.5). The internal and composite reliability, as well as the factor loading of the 21 items used to build the SEM (eg. *structure*, *process* and *outcome*), is presented in Table 2. Normality was checked again by testing the skewness and kurtosis. The test showed that the 21 items for the SEM were normally distributed.

As shown in Table 2, consumers generally reported being satisfied with the health systems *structure*: accessibility (Mean = 3.62; SD = 1.00), managing side effects (Mean = 3.87; SD = 0.95), response of service to crises (Mean = 3.8; SD = 0.84), listen to the worries of relatives (Mean = 3.8; SD = 0.88), recommendation made to relatives (Mean = 3.79; SD = 0.93) and information to relatives (Mean = 3.70; SD = 1.02). Conversely, consumers had mixed satisfaction regarding the affordability of services (Mean = 2.72; SD = 1.23).

Also, consumers generally reported being satisfied with the *process* indicators, such as competency of psychiatrist (Mean = 3.98; SD = 0.79), psychiatrist listening & understanding of illness (Mean = 4.09; SD = 0.85), instructions about appointment (Mean = 3.87; SD = 0.87), cooperation between service providers (Mean = 3.84; SD = 0.84), confidentiality and respect for your right (Mean = 4.07; SD = 0.81), information about diagnosis & prognosis (Mean = 3.57; SD = 1.15) as well as explanation of procedures & approaches (Mean = 3.57; SD = 0.97).

**Table 1. Characteristics of consumers.**

| Variables | Facility 1 | Facility 2 | Psy. Unit | Total |
|---|---|---|---|---|
| *Demographic information* | *N (%)* | *N (%)* | *N (%)* | *N (%)* |
| **Age**[*] | | | | |
| **18–27** | 28 (35.0) | 80 (29.1) | 61 (39.9) | 169 (33.3) |
| **28–37** | 16 (20.0) | 96 (34.9) | 67 (43.8) | 179 (35.2) |
| **38–47** | 9 (11.3) | 55 (20.0) | 19 (12.4) | 83 (16.3) |
| **48–57** | 17 (21.3) | 27 (9.8) | 5 (3.3) | 49 (9.7) |
| **$\geq$ 58 years** | 10 (12.5) | 17 (6.2) | 1 (0.7) | 28 (5.5) |
| **Gender** | | | | |
| **Male** | 36 (45.0) | 135 (48.7) | 92 (60.1) | 263 (51.6) |
| **Female** | 44 (55.0) | 142 (51.3) | 61 (39.9) | 247 (48.4) |
| **Education** | | | | |
| **None** | 5 (6.25) | 17 (6.14) | 2 (1.3) | 24 (4.7) |
| **Basic (Primary and JHS)** | 25 (31.25) | 63 (22.74) | 14 (9.15) | 102 (20.0) |
| **Senior High school/Voc./Tech.** | 31 (38.75) | 111 (40.1) | 63 (41.18) | 205 (40.2) |
| **Tertiary** | 19 (23.75) | 86 (31.1) | 74 (48.37) | 179 (35.1) |
| **Marital Status** | | | | |
| **Single** | 48 (60.0) | 188 (67.87) | 126 (82.4) | 362 (70.9) |
| **Married** | 20 (25.0) | 62 (22.4) | 22 (14.4) | 104 (20.4) |
| **Separated** | 6 (7.5) | 21 (7.58) | 4 (2.6) | 31 (6.08) |
| **Widow** | 6 (7.5) | 6 (2.17) | 1 (0.65) | 13 (2.6) |
| **Primary Occupation** | | | | |
| **Student** | 13 (16.25) | 51 (18.41) | 37 (24.2) | 101 (19.8) |
| **Unemployed** | 18 (22.50) | 70 (25.27) | 28 (18.3) | 116 (22.8) |
| **Skilled (eg, teaching, banking, nurse)** | 12 (15.0) | 50 (18.1) | 53 (34.6) | 115 (22.6) |
| **Semi (eg, artisan, mechanics)** | 15 (18.75) | 51 (18.4) | 28 (18.3) | 94 (18.4) |
| **Unskilled (eg, farmers, drivers)** | 22 (27.50) | 55 (19.9) | 7 (4.6) | 84 (16.5) |
| **Religion** | | | | |
| **Christianity** | 76 (95.0) | 251 (90.94) | 141 (92.2) | 468 (91.9) |
| **Islamic** | 3 (3.8) | 24 (8.7) | 8 (5.2) | 35 (6.9) |
| **Traditional** | 1 (1.25) | 0 | 1 (0.65) | 2 (0.4) |
| **Other** | 0 | 1 (0.36) | 3 (1.9) | 4 (0.8) |
| **NHIS Status** | | | | |
| **Insured (active)** | 50 (62.5) | 177 (63.9) | 122 (79.7) | 349 (68.4) |
| **Uninsured** | 30 (37.5) | 100 (36.1) | 31 (20.3) | 161 (31.6) |
| **Resident** | | | | |
| **Urban** | 55 (68.75) | 173 (62.45) | 101 (66.0) | 329 (64.5) |
| **Peri-Urban** | 16 (20.0) | 89 (32.13) | 34 (22.2) | 139 (27.3) |
| **Rural** | 9 (11.25) | 15 (5.42) | 18 (11.8) | 42 (8.24) |
| *Treatment pathways* | | | | |
| **Mental illness** | 21 (32.3) | 113 (45.8) | 73 (55.3) | 207 (46.6) |
| **Schizophrenia** | 18 (27.6) | 25 (10.1) | 30 (22.7) | 73 (16.4) |
| **Bipolar Affective Disorder Depression** | 10 (15.4) | 45 (18.2) | 19 (14.4) | 74 (16.7) |
| **Schizoaffective disorder** | 2 (3.1) | 11 (4.5) | 1 (0.76) | 14 (3.2) |
| **Substance-induced** | 4 (6.2) | 11 (4.5) | 1 (0.76) | 16 (3.6) |
| **Seizure disorder** | 5 (7.7) | 9 (3.6) | 3 (2.3) | 17 (3.8) |
| **Suicide Attempt** | - | 1 (0.40) | - | 1 (0.2) |
| **Anxiety disorder** | - | - | 2 (1.5) | 2 (0.5) |

*(Continued)*

**Table 1.** (Continued)

| Variables | Facility 1 | Facility 2 | Psy. Unit | Total |
|---|---|---|---|---|
| *Demographic information* | *N (%)* | *N (%)* | *N (%)* | *N (%)* |
| **Other neurological conditions** | 5 (7.7) | 13 (5.3) | - | 18 (4.1) |
| **Co-morbid conditions** | - | 19 (7.7) | 3 (2.3) | 22 (4.9) |
| **Mental health service** | | | | |
| **Outpatient service (visiting review)** | 73 (91.25) | 238 (85.92) | 122 (80.3) | 433 (85.1) |
| **Discharged from In-patient (0–12)** | 7 (8.75) | 39 (14.1) | 30 (19.7) | 76 (14.9) |
| **First point of care Visited the prayer centre/church** | 11 (13.75) | 59 (21.3) | 36 (23.5) | 106 (20.8) |
| **Visited herbalist** | 0 | 12 (4.3) | 3 (1.96) | 15 (2.9) |
| **Visited the imam/spiritual centre** | 0 | 1 (0.4) | 2 (1.3) | 3 (0.6) |
| **Visited the health facility** | 0 | 205 (74.0) | 112 (73.2) | 386 (75.7) |
| **Type of service provider** | | | | |
| **Health centre** | 3 (4.5) | 5 (2.4) | 5 (4.6) | 13 (3.39) |
| **Clinic** | 1 (1.45) | 6 (2.9) | 4 (3.6) | 11 (2.9) |
| **Hospital** | 15 (21.7) | 49 (23.9) | 61 (55.9) | 125 (32.6) |
| **Psychiatric hospital** | 50 (72.5) | 145 (70.7) | 39 (35.8) | 234 (61.1) |
| **Person who introduced a consumer** | | | | |
| **General practitioner** | 13 (19.1) | 31 (15.3) | 42 (38.5) | 86 (22.7) |
| **Community Mental health worker** | 0 | 1 (0.5) | 2 (1.8) | 3 (0.79) |
| **Pastor at the prayer centre** | 1 (1.5) | 1 (0.5) | 0 | 2 (0.53) |
| **Family member** | 52 (76.5) | 157 (77.7) | 58 (53.2) | 267 (70.5) |
| **Friends, colleague or community** | 2 (2.9) | 12 (5.9) | 7 (6.4) | 21 (5.5) |

*(Min; Max; Mean; SD) (18; 87; 34.46; 12.17)

Again, consumers generally reported being satisfied with *outcome* indicators such as attaining wellbeing & preventing relapse (Mean = 3.93; SD = 0.82), knowledge & understanding (Mean = 3.91; SD = 0.85), symptoms (Mean = 3.94; SD = 0.82), self-care (Mean = 3.98; SD = 0.87), relationship (Mean = 3.85; SD = 0.91), work/vocational skills (Mean = 3.81; SD = 0.93) and satisfaction (Mean = 3.96; SD = 0.72).

## Confirmatory factor analysis results

As shown in Table 3, the correlation matrix demonstrates sufficient convergent and discriminant correlation between the VSS-54 construct and WHODAS construct. Fig 1, graphically describes the hypothesized SEM. Results from the CFA showed that the hypothesized model had a good fit with the Residual Mean Square of Approximation (RMSEA) = 0.049; 90% CI, lower bound = 0.042; upper bound = 0.056; pclose = 0.596; Comparative fit index (CFI) = 0.951; Tucker-Lewis index (TLI) = 0.937; Coefficient of determination = 0.883. Details of the standardized coefficient estimates for CFA is represented in Fig 2 and Table 4. The path model result shows that the health systems *structure* was significantly related to the *process* and *outcome* ($\beta$ = 0.40; p<0.001). Similarly, the health systems *structure* construct was mediated by the *process* construct in its relationship with the *outcome* ($\beta$ = 0.346; p<0.001).

## Hypothesis test

**Hypothesis 1 (H$_1$): Process construct mediates the relationship between health systems structure and the outcome of mental health services.** To test for this mediating effect, the direct and indirect effect of the previously described path model from the CFA was examined.

**Table 2. Descriptive statistics of items.**

| Latent variable | Observed variables | Factor loading | Mean (SD) | Alpha |
|---|---|---|---|---|
| **Structure** | | | | |
| | Accessibility (A) | 0.38 | 3.62 (1.00) | 0.91 |
| | Affordability (B) | 0.39 | 2.72 (1.23) | 0.91 |
| | Managing Side effects (C) | 0.59 | 3.87 (0.95) | 0.90 |
| | Response of Service to Crises (D) | 0.60 | 3.8 (0.84) | 0.90 |
| | Listen to the worries of Relatives (E) | 0.77 | 3.8 (0.88) | 0.90 |
| | Recommendation to made to Relatives (F) | 0.76 | 3.79 (0.93) | 0.90 |
| | Information to Relatives (G) | 0.69 | 3.70 (1.02) | 0.90 |
| **Total** | | | 3.32 (0.80) | 0.75 |
| **Process** | Competency of psychiatrist (H) | 0.53 | 3.98 (0.79) | 0.90 |
| | Psychiatrist listen & understand illness (I) | 0.61 | 4.09 (0.85) | 0.90 |
| | Instructions about appointment (J) | 0.59 | 3.87 (0.87) | 0.90 |
| | Cooperation between service providers (K) | 0.55 | 3.84 (0.84) | 0.91 |
| | Confidentiality and respect for your right (L) | 0.63 | 4.07 (0.81) | 0.90 |
| | Information about diagnosis & prognosis (M) | 0.55 | 3.57 (1.15) | 0.90 |
| | Explanation of procedures & approaches (N) | 0.57 | 3.57 (0.97) | 0.90 |
| **Total** | | | 3.79 (0.64) | 0.80 |
| **Outcome** | | | | |
| | Attaining wellbeing &preventing relapse (O) | 0.71 | 3.93 (0.82) | 0.90 |
| | Knowledge & understanding (P) | 0.68 | 3.91 (0.85) | 0.90 |
| | Symptoms (Q) | 0.76 | 3.94 (0.82) | 0.90 |
| | Self-care (R) | 0.74 | 3.98 (0.87) | 0.90 |
| | Relationship (S) | 0.75 | 3.85 (0.91) | 0.90 |
| | Work/vocational skills (T) | 0.75 | 3.81 (0.93) | 0.90 |
| | Satisfaction (U) | 0.57 | 3.96 (0.72) | 0.90 |
| **Total** | | | 3.84 (0.70) | 0.88 |
| | | | | 0.91 |

As shown in Table 5, the health system *structure* had a positive and significant causal relationship with the dependant variable, *outcome* ($\beta = 0.47$; p<0.01). Similarly, health systems *structure* had a positive causal relationship with the mediator, *process* ($\beta = 0.60$; p<0.01). Again, the mediator (*process*) had a positive causal relationship with the *outcome* of mental health services ($\beta = 0.32$; p<0.01) (Fig 3). Finally, the indirect effects of health systems *structure* on the *outcome* as mediated by process was reduced and significant ($\beta = 0.19$; p<0.01).

Equation

1. Outcome = $0.32_{(process)}$ + $0.47_{(structure)}$ + $0.48_{(prediction\ error)}$, $R^2 = 0.46$

2. Process = $0.60_{(structure)}$ + $0.51_{(prediction\ error)}$, $R^2 = 0.36$

**Hypothesis 2 (H$_2$): Individual factors (e.g. gender, age, NHIS, resident and type of mental health services) moderates the relationship between the health systems structure and outcome of mental health services.** To test the interaction hypothesis, the factor scale score of the *outcome* was used as a dependant variable and the demographic characteristics (gender, age, and NHIS, resident) and treatment pathways (the type of services and first point of care) as independent variables. An interaction variable was generated by multiplying health systems *structure* and the independent variables. As shown in Table 6, none of the interactions was

**Table 3. Construct validity (convergent and divergent) of quality of mental health instrument (correlation for CFA and SEM analysis).**

| Dimensions | Satisfaction | Professional skills | Information | Access | Efficacy | Types of Intervention | Relative's Involvement | Cognition | Mobility | Self-care | Getting along | Life activities | Participation |
|---|---|---|---|---|---|---|---|---|---|---|---|---|---|
| Overall satisfaction | 1.00 | | | | | | | | | | | | |
| Professionals' Skills | 0.25** | 1.00 | | | | | | | | | | | |
| Information | 0.26** | 0.39** | 1.00 | | | | | | | | | | |
| Access | 0.12** | 0.29** | 0.34** | 1.00 | | | | | | | | | |
| Efficacy | 0.41** | 0.47** | 0.50* | 0.34** | 1.00 | | | | | | | | |
| Intervention | 0.29* | 0.11** | 0.27** | 0.24** | 0.33** | 1.00 | | | | | | | |
| Relative's Involvement | 0.28** | 0.37** | 0.50** | 0.22** | 0.49** | 0.28** | 1.00 | | | | | | |
| Cognition | -0.03 | -0.10 | -0.08* | -0.06 | -0.18** | -0.02 | -0.20** | 1.00 | | | | | |
| Mobility | 0.02 | -0.08 | 0.006 | -0.03 | -0.06 | 0.08 | -0.09* | 0.68** | 1.00 | | | | |
| Self-care | -0.01 | -0.15** | -0.05 | -0.03 | -0.14** | 0.07 | -0.11* | 0.65** | 0.73** | 1.00 | | | |
| Getting along | -0.05 | -0.16** | -0.09* | -0.08* | -0.19** | -0.003 | -0.16** | 0.68** | 0.63** | 0.76** | 1.00 | | |
| Life activities | -0.02 | -0.12** | -0.07 | -0.04 | -0.16** | 0.002 | -0.15** | 0.66** | 0.65** | 0.76** | 0.77** | 1.00 | |
| Participation | 0.04 | -0.11** | -0.11** | -0.09* | -0.18** | 0.007 | -0.15** | 0.59** | 0.58** | 0.65** | 0.72** | 0.77** | 1.00 |

**Significance $p < 0.01$;

*$p < 0.05$

significant except age. The analysis demonstrated that age had a significant and negative ($\beta = -0.014$; $p < 0.01$) moderating effect on the relationship between health systems *structure* and *outcome*. Contrarily, the insurance status ($\beta = 0.07$; $p > 0.05$) and type of services ($\beta = .025$; $p > 0.05$), i.e outpatient versus inpatient, had a positive moderating effect on the relationship between health systems *structure* and *outcome* but were not significant. It was also observed that the model had a good fit for each of the moderated independent variables.

## Discussion

This quantitative study assessed the mediation and moderation effects of health systems *structure* and *process* on the *outcome* of mental health services in Ghana. The findings from this study are discussed according to 1) the mediating effects of health systems *structure* and *process* on the *outcome* of services, and 2) the moderating effects of individual factors on the outcome of services.

### Mediation effects of health systems structure and process on the outcome of services

The results indicated that the consumers were mostly satisfied with the health system *structure*, *process* and *outcome* of mental health services. Based on our measures, this suggests that the consumers were satisfied with the accessibility, adequacy, and availability of services. Similarly, consumers were mostly satisfied with the processes measures during their treatment, including professional competency and therapeutic relationships. Also, the findings demonstrated that consumers were mostly satisfied with the outcome of treatment, particularly on the attainment of well-being and relapse prevention, gaining knowledge and understanding of illness, improvement in symptoms, self-care, getting along with people, vocational skills and general satisfaction with services.

Although consumers were mostly satisfied with the indicators for structure, process and outcome, they generally had a mixed satisfaction with the affordability of the services.

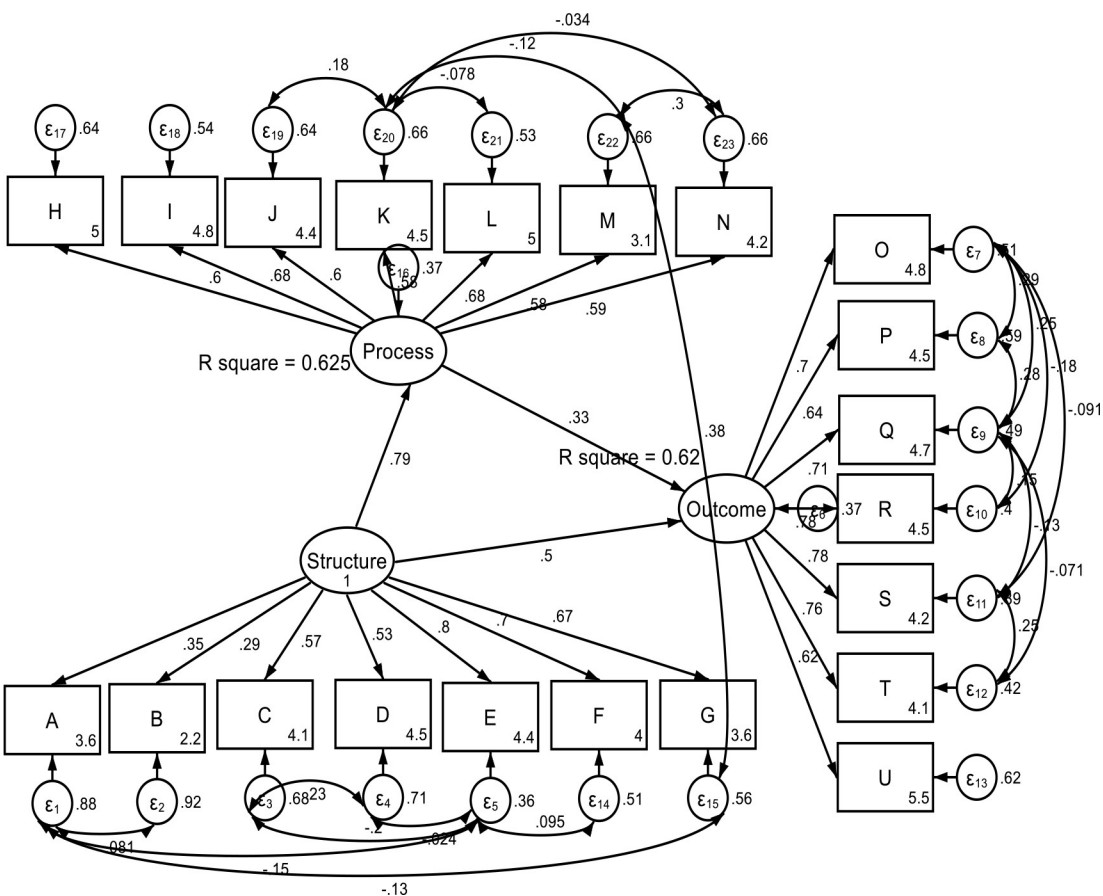

**Fig 2. Standardized coefficient estimates of hypothesized SEM.**

Consistent with previous studies [49, 50], consumers generally faced challenges in financing their mental health treatment. This could be attributed to the existing financing mechanism of mental health services. Whilst a national health insurance policy has been implemented over the past decade in Ghana as part of the government effort to remove out-of-pocket payment, mental health services are not yet covered by such policies. The existing government support for mental health services is largely directed towards the management of psychiatric facilities, and some contribution towards policy and governance, with almost no funding to support direct payment for services [49, 50]. Capacity to pay for mental health services for consumers comes from personal sources, family caregivers, as well as individual benevolent support. These sources, however, are typically inadequate to meet the needs of consumers.

The study also found that the mental health system *structure* and *process* had a direct causal relationship with the *outcome* of services. This implies that the impact of the health system on the outcome of mental health services was mediated by the process, which reflects the technical competency of mental health professionals, as well as the therapeutic relationship. These findings extend our understanding of quality assessment in mental health by supporting previous Donabedian theory [22, 23, 51–53] as well as recent conceptual frameworks for measuring the quality of mental health services [2]. Previous theories have suggested that the structure and organisation of the health system has a significant role in improving the mental health outcomes of consumers [22, 23, 51–53]. The findings indicate that strengthening the health system through adequate financing or insurance policies, trained human resources as well as the

**Table 4. Standardized coefficient estimates for CFA.**

| Variables | Latent constructs | β | SE | 95% CI |
|---|---|---|---|---|
| *Structural* | | | | |
| Process | Outcome | 0.40** | 0.10 | 0.18–0.61 |
| Structure | Outcome | 0.81** | 0.18 | 0.45–1.18 |
| Structure | Process | 1.06** | 0.16 | 0.73–1.39 |
| *Measurement* | | | | |
| Accessibility | Structure | 1.00 | | |
| Affordability | Structure | 0.99** | 0.20 | 0.60–1.39 |
| Managing Side effects | Structure | 1.51** | 0.24 | 1.04–1.98 |
| Response of Service to Crises | Structure | 1.29** | 0.21 | 0.87–1.70 |
| Listen to the worries of Relatives | Structure | 2.00** | 0.29 | 1.42–2.59 |
| Recommendation to relatives | Structure | 1.85** | 0.27 | 1.31–2.40 |
| Information to Relatives | Structure | 1.92** | 0.29 | 1.34–2.50 |
| Competency of psychiatrist | Process | 1.00 | | |
| Psychiatrist listen & understand illness | Process | 1.21** | 0.10 | 1.00–1.41 |
| Instructions about appointment | Process | 1.11** | 0.10 | 0.90–1.32 |
| Cooperation between service providers | Process | 1.03** | 0.10 | 0.82–1.25 |
| Confidentiality and respect for your right | Process | 1.17** | 0.10 | 0.97–1.38 |
| Information about diagnosis and prognosis | Process | 1.41** | 0.13 | 1.13–1.68 |
| Explanation of procedures and approaches | Process | 1.12** | 0.11 | 0.90–1.33 |
| Attaining wellbeing | Outcome | 1.00 | | |
| Knowledge & understanding | Outcome | 0.95** | 0.06 | 0.82–1.08 |
| Symptoms | Outcome | 1.02** | 0.06 | 0.89–1.16 |
| Self-care | Outcome | 1.19** | 0.08 | 1.02–1.36 |
| Relationship | Outcome | 1.24** | 0.08 | 1.07–1.42 |
| Work/vocational skills | Outcome | 1.24** | 0.08 | 1.06–1.42 |
| Satisfaction | Outcome | 0.77** | 0.06 | 0.64–0.89 |

**Significance p<0.01; Goodness of Fit: RMSEA = 0.049; 90% CI, lower bound = 0.042; upper bound = 0.056; pclose = 0.596 Probability RMSEA < = 0.05; Comparative fit index (CFI) = 0.951; Tucker-Lewis index (TLI) = 0.937; Coefficient of determination = 0.883

**Table 5. Path analysis of mediation effects of structure and process on the outcome.**

| Model | β | SE | 95% CI | Hypotheses |
|---|---|---|---|---|
| *Direct effects* | | | | |
| Process → outcome | 0.32** | .05 | 0.22–0.43 | Yes |
| Structure → outcome | 0.47** | 0.05 | 0.36–0.57 | Yes |
| Structure → process | 0.60** | 0.04 | .52 –.69 | Yes |
| *Indirect effects* | | | | |
| Structure → outcome | 0.19** | 0.03 | 0.12–0.26 | Yes |
| *Total effects* | | | | |
| Process → outcome | 0.32** | 0.05 | .22 –.43 | Yes |
| Structure → outcome | 0.66** | 0.04 | .58 –.75 | Yes |
| Structure → process | 0.60** | 0.04 | .52 –.69 | Yes |

**Significance p<0.01; Goodness of Fit: RMSEA = 0.000; 90% CI, lower bound = 0.000; upper bound = 0.000; pclose = 1.000 Probability RMSEA < = 0.05; Comparative fit index (CFI) = 1.000; Tucker-Lewis index (TLI) = 1.000; Coefficient of determination = 0.48

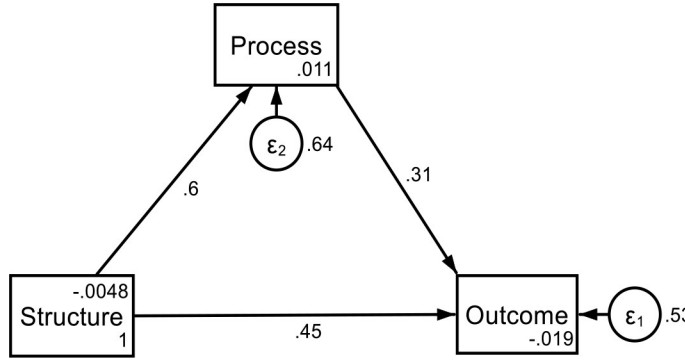

**Fig 3. Pathway analysis diagram of the mediation effect.**

efficient availability of outpatient and inpatient services for all consumers, is likely to improve consumer outcomes for the mental health services provided [11, 12]. This finding aligns with previous research, for example, Kunkel, Rosenqvist [11] who found a significant correlation between the health systems structure and process indicators on the outcome of mental health services in Sweden. Similarly, Boden, Smith [12] concluded that mental health staffing ratios had substantial positive relationships with overall mental health treatment access and quality in the USA. Although previous studies in Sweden and USA have emphasised the positive impact of the health systems structure on outcome of mental health services, they were limited to the perspectives of mental health professionals, without input from consumers. The current results provide a significant contribution as they reflect the consumers' perspective, therefore highlighting that the health system structure should be a primary consideration when developing measures to improve the outcome of mental health services. This could be realised through a greater emphasis on effective monitoring and a deeper evaluation of mental health system performance, as well as providing adequate organizational and consumer support services.

## The moderation effects of individual factors on the outcome of services

The study found that the relationship between the health system structure and the outcome of mental health services is moderated by individual consumer factors including health insurance

**Table 6. Moderating effects of demographic characteristics and treatment pathways on the outcome.**

| Variables | β | SE | 95% CI |
|---|---|---|---|
| Gender | 0.04[(ns)] | 0.06 | -0.08–0.17 |
| Gender as moderator | -0.16[(ns)] | 0.08 | -0.33–0.008 |
| Age | 0.003[(ns)] | 0.002 | -.002–0.008 |
| Age as moderator | -0.014** | 0.003 | -0.02 –(-0.006) |
| Insurance | 0.039[(ns)] | 0.07 | -0.10–0.18 |
| Insurance as moderator | 0.07[(ns)] | 0.10 | -0.13–0.28 |
| Resident | -0.08[(ns)] | .06 | -0.22–0.05 |
| Type of services | 0.11[(ns)] | 0.10 | -0.08–0.31 |
| Type of service as moderator | .025[(ns)] | 0.14 | -0.26–0.31 |
| Treatment pathways | -0.01[(ns)] | 0.078 | -0.17–0.13 |
| Treatment pathways as moderator | -0.09[(ns)] | 0.098 | -0.28–0.10 |

[ns]Not significant

**Significance $p < 0.01$

status and the type of services. Although the impact of these individual factors were not statistically significant, they had a positive moderating effect on the relationship between health systems *structure* and *outcome*. As discussed above, the individual factors confirm previous Donabedian and Thomas and Pechansky theory's on access and quality of mental health services [22, 23, 51, 52, 54].

Further, the age of consumers had a negative significant moderating effect on the relationship between health system *structure* and *outcome* of mental health services. This finding suggests that as the age of consumers reduces the effects of health systems *structure* on the outcome of services increases. Conversely, increased age of consumers significantly reduces the effects of the positive relationship between health systems *structure* and outcome of services. The findings indicated that younger age was associated with increased access and satisfaction with the quality of mental health services. Previous findings from developed countries have yielded ambiguous results regarding the relationship between age and the quality of mental health services. For instance, whilst some studies concluded that adult age is inversely associated with increased satisfaction and access to services [55], others had contrary findings [56, 57]. The differences in evidence from previous studies in developed countries could be attributed to the differences in the health care systems and individual demographics of the population. For instance, developed countries may have better resources and social services, compared with developing countries.

### Limitations

The study has some limitations that need recognition when interpreting the findings. The limitations were associated with the vulnerability of consumers when using RedCap. For instance, consumers receiving mental health services may have experienced negative affective reaction when completing survey instruments. Further, the nature of the survey may have led some participants to experience psychological and emotional distress. However, we attempted to attenuate these risks by, for example, ensuring that participants gave informed consent, pre-testing survey tools, as well as monitoring of the data collection process helped to achieve the highest reliability, validity and timeliness of data collection.

### Conclusion

The study concludes that consumers were mostly satisfied with their experience of health system *structure*, *process* as well as the *outcome* of mental health services. Despite this, consumers had a mixed reaction or reduced satisfaction with the affordability of the services provided. The study also concludes that the health systems *structure* and the *process* had a significant impact on the outcome of mental health services. More specifically, the health system *structure* and *process* had a direct positive causal relationship with the outcome of mental health services. The effects of the health system *structure* on the *outcome* of mental health services were mediated by the process, reflecting technical competency of MHPs and therapeutic relationships. Moreover, the individual consumer factors, such as age, NHIS, type of service accessed moderated the relationship between the health systems *structure* and the *outcome* of mental health services.

### Implications for policy and mental health practice

The study has several important implications for policy and mental health clinical practice. Given that consumers had mixed satisfaction regarding the affordability of services, we recommend that the government through the mental health authority should revise the existing national health insurance scheme to include mental health services. The cost of mental health

services should be fully financed by the NHIS to reduce the out-of-pocket payment, as in some developed countries. Similarly, it is proposed that the Ghana government should strengthen the mental health system to attain an improved outcome for consumers. This could be achieved through effective monitoring and evaluation of mental health systems performance, as well as providing adequate organizational and consumer support services. Our findings also recommend that government mental health policy could provide additional support for older people with mental illness. Specifically, mental health professionals are encouraged to consider consumer age when planning and monitoring mental health services. Such measures could help to support the mental health needs of the aged population.

The study also suggests that future studies should investigate approaches to financing mental health services through, for example, insurance policies, particularly relating to the outcomes of services. Further, research could also attempt to develop a user-friendly and context-specific measurement scale that is specific to LMICs setting, on health system *structure* and *process* indicators that could achieve better consumer outcomes. Finally, future research should aim to use a qualitative method to explore the subjective experiences of older people in mental health service provision. Specifically, such studies could explore the reasons why older consumers are more likely to experience poor quality mental health services.

## Supporting information

**S1 Survey. The quality of mental health services in Ghana: Providers and consumers perspectives.**
(PDF)

## Acknowledgments

The authors are also grateful with the support received from the staff of Pantang Psychiatric Hospital, Accra Psychiatric Hospital and Psychiatric Unit of the Komfo Anokye Teaching Hospital for supporting the data collection.

## Author Contributions

**Conceptualization:** Eric Badu, Anthony Paul O'Brien, Rebecca Mitchell, Akwasi Osei.

**Data curation:** Eric Badu, Anthony Paul O'Brien.

**Formal analysis:** Eric Badu, Anthony Paul O'Brien, Rebecca Mitchell.

**Investigation:** Eric Badu.

**Methodology:** Eric Badu, Anthony Paul O'Brien, Rebecca Mitchell, Akwasi Osei.

**Project administration:** Eric Badu, Anthony Paul O'Brien, Rebecca Mitchell, Akwasi Osei.

**Supervision:** Eric Badu, Anthony Paul O'Brien, Rebecca Mitchell, Akwasi Osei.

**Writing – original draft:** Eric Badu, Anthony Paul O'Brien, Rebecca Mitchell.

**Writing – review & editing:** Eric Badu, Anthony Paul O'Brien, Rebecca Mitchell, Akwasi Osei.

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
