## [Decision Letter · Decision Letter 0]

3 Apr 2020

PONE-D-20-07739

Mediating and moderating effects of structure and process on the quality of mental health services – structural equation modelling

PLOS ONE

Dear Mr. Badu,

Thank you for submitting your manuscript to PLOS ONE. After careful consideration, we feel that it has merit but does not fully meet PLOS ONE’s publication criteria as it currently stands. Therefore, we invite you to submit a revised version of the manuscript that addresses the points raised during the review process.

We would appreciate receiving your revised manuscript by May 18 2020 11:59PM. To enhance the reproducibility of your results, we recommend that if applicable you deposit your laboratory protocols in protocols.io, where a protocol can be assigned its own identifier (DOI) such that it can be cited independently in the future. For instructions see: http://journals.plos.org/plosone/s/submission-guidelines#loc-laboratory-protocols

We look forward to receiving your revised manuscript.

Kind regards,

Lars-Peter Kamolz, M.D., Ph.D., M.Sc.

Academic Editor

PLOS ONE

Journal Requirements:

3. Please include additional information regarding the survey or questionnaire used in the study and ensure that you have provided sufficient details that others could replicate the analyses. For instance, if you developed a questionnaire as part of this study and it is not under a copyright more restrictive than CC-BY, please include a copy, in both the original language and English, as Supporting Information. Moreover, please include more details on how the questionnaire was pre-tested, and whether it was validated.

4. Please ensure that you refer to Figure 1 in your text as, if accepted, production will need this reference to link the reader to the figure.

Reviewers' comments:

Reviewer's Responses to Questions

**Comments to the Author**

1. Is the manuscript technically sound, and do the data support the conclusions?

Reviewer #1: Partly

Reviewer #2: Yes

2. Has the statistical analysis been performed appropriately and rigorously? 

Reviewer #1: Yes

Reviewer #2: Yes

3. Have the authors made all data underlying the findings in their manuscript fully available?

Reviewer #1: Yes

Reviewer #2: Yes

4. Is the manuscript presented in an intelligible fashion and written in standard English?

Reviewer #1: Yes

Reviewer #2: Yes

5. Review Comments to the Author

Reviewer #1: Dear authors,

I got to review your manuscript "Mediating and moderating effects of structure and process on the quality of mental health services - structural equation modelling.", in which you present the results of a survey of 510 mental health consumers in Ghana considering their satisfaction with the services.

Before considering publication, I recommend to improve the manuscript concerning the following:

1.) Since three of the four authors are from Australia (or at least affiliated), it was quite surprising that the study was in set in establishments in Ghana; I therefore recommend to already mention that in the title.

2.) In my opinion, there is a lot of superfluous information in the introduction, which is not really to the point. There are some aspects, that would rather fit in the discussion (like e.g. lines 105 to 113). I suggest to significantly shorten the introduction, focus a little more on the actually relevant aspects and clearly present the purpose of the research, which was quite hard to grasp while reading the introduction.

3.) Please revise the manuscript concerning grammar and orthography, since there are a lot of mistakes and inattentivenesses (like wrongly placed parentheses).

4.) Please also revise the wording and structure of the manuscript, as there are many repetitions and the manuscript is quite hard to read with an interrupted reading flow - odd wordings also contribute to this (e.g. lines 318f make no sense in the current wording). It appears more like a listing of facts (or results), than a comprehensive text.

5.) The percentages in table 1 do not add up to 100% (probably just rounding errors). Please correct this.

6.) Please also consider revising the discussion section and try writing a little more structured. While reading, I got the impression, that there were a lot of repetitions (regarding content and wording). Again, it is not to the point.

Thank you.

Reviewer #2: Dear authors

Thank you for the opportunity to review the mansucript, „Mediating and moderating effects of structure and process on the quality of mental health services – structural equation modelling.

In principle, an absolutely interesting manuscript, and its publication is desirable.

The manuscript is, in principle, a well-structured/logically structured study on the basis of an excellent data pool.

A few minute remarks:

The Introduction & Discussion Section should be written more stringently and shorter.

6. PLOS authors have the option to publish the peer review history of their article (what does this mean?). If published, this will include your full peer review and any attached files.

Reviewer #1: No

Reviewer #2: No

---

## [Author Response · Author response to Decision Letter 0]

20 Apr 2020

April 14, 2020

Dear Editor,

Response to Reviewers Query regarding the manuscript: “Mediation and moderation effects of health systems structure and process on the quality of mental health services in Ghana – structural equation modelling.” 

Thank you for the opportunity to submit the above manuscript for consideration in your reputable journal. Please I write on behalf of the authors to submit a response to the queries provided in the above manuscript. The authors have responded to all the reviewers' queries and highlighted them in yellow ink in the revised version of the manuscript. The reviewer's query has been addressed as follows:

Reviewer 1 Query

1.) Since three of the four authors are from Australia (or at least affiliated), it was quite surprising that the study was in set in establishments in Ghana; I therefore recommend to already mention that in the title.

Authors response

Thank you for the comments. The authors have included “Ghana” in the title of the paper

Reviewer 1 Query

2.) In my opinion, there is a lot of superfluous information in the introduction, which is not really to the point. There are some aspects that would rather fit in the discussion (like e.g. lines 105 to 113). I suggest to significantly shorten the introduction, focus a little more on the actually relevant aspects and clearly present the purpose of the research, which was quite hard to grasp while reading the introduction.

Authors Response

Thank you for the suggestion. The authors have shorten the introduction section particularly regarding the last two paragraphs. The authors have clearly justified the gaps in literature and the purpose of the study. 

Reviewer 1 Query

3.) Please revise the manuscript concerning grammar and orthography, since there are a lot of mistakes and in attentiveness (like wrongly placed parentheses).

Authors Response

In relation to the reviewers query, the authors have revised the entire manuscript for clarity regarding grammar and orthography. Thank you for this comment

Reviewer 1 Query

4.) Please also revise the wording and structure of the manuscript, as there are many repetitions and the manuscript is quite hard to read with an interrupted reading flow - odd wordings also contribute to this (e.g. lines 318f make no sense in the current wording). It appears more like a listing of facts (or results), than a comprehensive text.

Authors Response

Thank you for the comments. The authors have revised the wording and structure of the manuscript. In particular, the authors have shorten the interpretation of the results

Reviewer 1 Query

5.) The percentages in table 1 do not add up to 100% (probably just rounding errors). Please correct this.

Authors Response

Thank you for drawing our attention to this matter. The authors have revised the section. The authors wish to clarify that the summation is based on the column total for each variable disaggregated by facility (Facility 1, Facility 2 and Psychiatric Unit). 

Reviewer 1 Query

6.) Please also consider revising the discussion section and try writing a little more structured. While reading, I got the impression, that there were a lot of repetitions (regarding content and wording). Again, it is not to the point.

Authors Response

Thank you for highlighting this issue. The authors have revised the discussion to reduce duplication and increase the clarity of our writing

Reviewer 2 Query

The Introduction & Discussion Section should be written more stringently and shorter.

Authors response

Thank you for the suggestion. The authors have revised the introduction and discussion. In particular, we have reduced duplication as well as word count. We have undertaken a thorough review of our writing to enhance readability. All authors have edited the manuscript thoroughly for grammatical errors.

---

## [Decision Letter · Decision Letter 1]

5 May 2020

Mediation and moderation effects of health systems structure and process on the quality of mental health services in Ghana – structural equation modelling.

PONE-D-20-07739R1

Dear Dr. Badu,

We are pleased to inform you that your manuscript has been judged scientifically suitable for publication and will be formally accepted for publication once it complies with all outstanding technical requirements.

With kind regards,

Lars-Peter Kamolz, M.D., Ph.D., M.Sc.

Academic Editor

PLOS ONE

Additional Editor Comments (optional):

Reviewers' comments:

Reviewer's Responses to Questions

**Comments to the Author**

1. If the authors have adequately addressed your comments raised in a previous round of review and you feel that this manuscript is now acceptable for publication, you may indicate that here to bypass the “Comments to the Author” section, enter your conflict of interest statement in the “Confidential to Editor” section, and submit your "Accept" recommendation.

Reviewer #1: All comments have been addressed

Reviewer #2: All comments have been addressed

---

## [Editor Report · Acceptance letter]

11 May 2020

PONE-D-20-07739R1 

Mediation and moderation effects of health system structure and process on the quality of mental health services in Ghana – structural equation modelling. 

Dear Dr. Badu:

I am pleased to inform you that your manuscript has been deemed suitable for publication in PLOS ONE. Congratulations! Your manuscript is now with our production department. 

With kind regards,

on behalf of

Dr. Lars-Peter Kamolz 

Academic Editor

PLOS ONE